# Analysis of Mycotoxin and Secondary Metabolites in Commercial and Traditional Slovak Cheese Samples

**DOI:** 10.3390/toxins14020134

**Published:** 2022-02-10

**Authors:** Luana Izzo, Petra Mikušová, Sonia Lombardi, Michael Sulyok, Alberto Ritieni

**Affiliations:** 1Department of Pharmacy, University of Naples “Federico II”, Via Domenico Montesano 49, 80131 Naples, Italy; sonia.lombardi@unina.it (S.L.); alberto.ritieni@unina.it (A.R.); 2Department of Cryptogams, Plant Science and Biodiversity Center, Institute of Botany, Slovak Academy of Sciences, Dúbravská Cesta 9, SK-84523 Bratislava, Slovakia; petra.mikusova@savba.sk; 3Department of Agrobiotechnology (IFA-Tulln), Institute of Bioanalytics and Agro-Metabolomics, University of Natural Resources and Life Sciences Vienna, Konrad Lorenzstr. 20, A-3430 Tulln, Austria; michael.sulyok@boku.ac.at; 4Health Education and Sustainable Development, University of Naples “Federico II”, 80131 Naples, Italy

**Keywords:** mycotoxins, Slovak cheeses, fungi growth, enniatin B, tryptophol

## Abstract

Cheese represents a dairy product extremely inclined to fungal growth and mycotoxin production. The growth of fungi belonging to *Aspergillus*, *Penicillium*, *Fusarium*, *Claviceps*, *Alternaria*, and *Trichoderma* genera in or on cheese leads to undesirable changes able to affect the quality of the final products. In the present investigation, a total of 68 types of commercial and traditional Slovak cheeses were analyzed to investigate the occurrence of fungal metabolites. Altogether, 13 fungal metabolites were identified and quantified. Aflatoxin M1, the only mycotoxin regulated in milk and dairy products, was not detected in any case. However, the presence of metabolites that have never been reported in cheeses, such as tryptophol at a maximum concentration level from 13.4 to 7930 µg/kg (average: 490 µg/kg), was recorded. Out of all detected metabolites, enniatin B represents the most frequently detected mycotoxin (0.06–0.71 µg/kg) in the analyzed samples. Attention is drawn to the lack of data on mycotoxins’ origin from Slovak cheeses; in fact, this is the first reported investigation. Our results indicate the presence of fungal mycotoxin contamination for which maximum permissible levels are not established, highlighting the importance of monitoring the source and producers of contamination in order to protect consumers’ health.

## 1. Introduction

Fungi are the major producers of secondary toxic metabolites, which represent the main cause of food spoilage. Mycotoxins are toxic compounds produced by several molds belonging to the main genera of fungi—*Aspergillus*, *Penicillium*, *Fusarium*, *Claviceps*, *Alternaria*, *Stachybotrys* and many others—which have a wide variety of properties that influence our common life [1]. The occurrence of toxic metabolites in food, as well as in feed, represents a serious global problem for human health due to their toxic effects, e.g., neurotoxic, nephrotoxic, carcinogenic and mutagenic effects [2]. Concerning food safety, European institutions established guidelines to minimize human exposure to different mycotoxins, e.g., to aflatoxins, ochratoxins, patulin, fumonisins and others, where their occurrence and maximum levels have been established [3,4].

Milk can contain various mycotoxins when lactating animals ingest contaminated feeds. Mycotoxins are metabolized, biotransformed, and transferred to animal products (milk, meat). There are mycotoxins of primary concern detected in dairy products and also mycotoxins that can be found as a result of in situ production by potentially toxigenic fungi frequently associated with cheese [5]. In fact, rumen flora can change a number of mycotoxins into metabolites that are less biologically inactive at common exposure levels. However, this does not apply to all mycotoxins that contaminate feed materials [6]. Milk and dairy product contamination may arise from several causes, such as animal feed contamination, starter strains, environment, processing equipment or incorrect manipulation [7]. The preparation and storage conditions of animal feedstuffs (grain, silage) can reconcile mycotoxin contamination. Toxic fungal metabolites can be produced and excreted by toxigenic species growing on cheese and penetrating the product [8]. It is concluded that the penetration depends on the type of cheese and the type of mycotoxin. However, it has also been reported by several authors that toxin concentrations and visible mold colonies may not always correlate [9,10,11,12].

The most common molds that contaminate cheese are *Penicillium* species [13], which can grow even at refrigerator temperature and produce the toxins ochratoxin A, citrinin, penicillic acid, patulin, mycophenolic acid, penitrem A, and cyclopiazonic acid [1]. The most frequently and commonly dominant *Penicillium* species on spoiled cheeses from different countries were identified. *P. commune* was the most widespread and most frequently occurring species. Most of the isolates belonged to the following species: *P. commune*, *P. nalgiovense*, *P. verrucosum*, *P. solitum*, *P. roqueforti*, *A. versicolor*, *P. crustosum*, *P. atramentosum*, *P. chrysogenum* and *P. echinulatum* [14,15]. Some of them, especially *P. commune*, *P. verrucosum*, *A. versicolor* or *P. Roqueforti,* are species reported as players in cheese ripening microbiota [16]. Moreover, a new *Penicillium* species, *P. gravini caseii,* was isolated and described from cave cheese [17]. As reported by Kandasamy et al. [18], that investigated ten dairy farms from six different provinces in the Republic of Korea, mycological analysis in cheese factories was important in an attempt to prevent mold growth on cheese. From this study emerged fungal contamination in 8 out of 10 dairy farms which were out of the acceptable range, as per hazard analysis critical control point regulation. The environment of cheese ripening rooms persuades a favorable niche for mold growth. The proper management of hygienic and production practices, and air filtration systems, would be effective to eradicate contamination in cheese processing industries [18]. 

To date, EU General Food Law, and other worldwide organizations, do not regulate the mycotoxins in cheese. In the European Union (EU), AFM1 is the only regulated mycotoxin in raw milk, heat-treated milk and milk for the manufacture of dairy products, at a concentration level not exceeding 0.05 μg/kg for adult consumption, and 0.025 μg/kg for food products meant for infants and young children [4]. Although AFM1 is the only mycotoxin regulated by the EU, the list of detected mycotoxins in cheeses is more extensive. The occurrence of ochratoxin A, patulin and citrinin [15,17,19,20], cyclopiazonic acid [21,22], roquefortine C and mycophenolic acid (ROQC) [23,24], isofumigaclavines [25], penitrems [26] or andrastins [27] were reported in the literature.

Thanks to the development of specific and sensitive tools for quantitative assessment of contaminants in foods, we are witnessing increasing work on this topic. Recent advances in mycotoxin quantifications have been focused on analytical methods for multi-mycotoxin analysis with minimal sample clean-up adapted to different types of food [28,29,30,31,32]. Until now, few scientific works have focused on the investigation of mycotoxins on Slovak cheeses. The occurrence and diversity of yeasts and filamentous fungi in Brydza cheeses were investigated by [33,34]. Cisárová et al. [35] studied the ability to produce cyclopiazonic acid by strains of fungi from Camembert-type cheese from different European countries, including Slovakia. 

Based on the above, the present scientific work is focused on the analysis, for the first time, of mycotoxins and secondary metabolites in 68 types of commercial and traditional Slovak cheeses through LC-ESI-MS/MS analysis. 

## 2. Results and Discussions

### 2.1. Mycotoxins and Metabolites Detected in Slovak Cheeses

From altogether 68 analyzed Slovak cheese samples, 13 different fungal metabolites in different concentrations were detected. Of all the tested samples, four samples (samples: 4, 6, 30 and 37) reported contamination from nearly all the tested mycotoxins, including samples 30 and 37, derived from commercial sheep cheeses, not smoked and smoked, respectively. Results obtained from original traditional Slovak cheeses came from markets and local farmers using a traditional biotechnical process at manufacture. Instead, the occurrence of target compounds investigated in common commercial cheeses that came from markets are reported in Appendix A. A summary of the analyzed Slovak cheeses divided from milk origin and their concentrations range are shown in Table 1. 

Among 28 target mycotoxins and metabolites investigated in analyzed samples, AFM1, the only mycotoxin regulated in milk and dairy products, was not detected in any case, which highlighted the overall milk quality (goat’s, cow’s, sheep’s) used for the cheese manufacturing. 

The complex diet of ruminants, consisting of forages, concentrates, and preserved feeds, can be a source of very diverse mycotoxins that contaminate individual feed components. In a brief review [36] was reported the assessment of mycotoxins in the diet of dairy cows in terms of exposure assessment. A direct consequence of the complex and variable composition of ruminant diets is the risk of exposure to more than one mycotoxin or a set of mycotoxin clusters produced by an individual fungal species. This includes, for example, aflatoxins, fumonisins, zearalenone, trichothecenes and ergot alkaloids; from pasture grasses it can be a source of lolitrems, paspalitrems, penitrem A, ergovaline and associated ergot alkaloids; and finally, contamination from preserved feeds (silage) where we talk about patulin, mycophenolic acid, roquefortines, fumitremorgens, verruculogen, monacolines, and others.

#### 2.1.1. Enniatin B

Out of all detected metabolites, enniatin B (ENN B) represents the only one detected in all analyzed samples, as well as in commercial and traditional cheeses. Although present in all samples (100%, *n* = 68), its concentration was low compared with other metabolites. ENN B concentrations ranged from under 0.01 to 0.71 µg/kg. Particularly, traditional cheeses were contaminated with slightly higher average levels compared with commercial cheeses: 0.21 and 0.15 µg/kg, respectively (Appendix A, Table 1).

ENNs belong to the so-called “emerging mycotoxins” family because of their increase in food and feed, and also the great concern about their worldwide presence, originally reported as being produced by *Fusarium* spp. [37,38]. ENNs occur as contaminants mainly in cereals, although their presence has been also reported in other matrices, including products of animal origin, as a consequence of the carry-over of these compounds into animal tissues after the feeding of contaminated feed [39,40,41,42,43,44,45]. As reported, high contamination levels of ENNs in food commodities have been reported. ENNs B, B1, B4, A, and A1 were quantified in baby food, including infant formula kinds of milk, dairy products such as cheese and yogurt, cereal-based baby food, fruit and vegetable compotes, fruits, and vegetable puree from the Italian market. The detected concentrations ranged from 11.8 to 832 µg/kg and the highest reported contamination levels were represented by ENN B [46]. In particular, ENN B is the most prevalent compound belonging to this group of mycotoxins and its presence represents relatively high concentrations in *Fusarium*-contaminated food and feed. In a multi-mycotoxin validation of an efficient multi-analyte method for the detection and quantification of mycotoxins of maize silage from Spain dairy farms, Dagnac et al. [47] found that ENN B was the most frequent mycotoxin; it achieved the highest average detection frequency and was detected in 51% of the samples (*n* = 148; average concentration: 157 µg/kg). In addition, ENN B was confirmed by a pilot study for the presence of fungal metabolites in sheep milk from first spring milking. Out of 700 bacterial, fungal and plant metabolites tested for, only one mycotoxin—ENN B—was detected in sheep milk samples (0.0055–0.0121 μg/kg; average concentration: 0.0078 μg/kg) [48]. Unlike other *Fusarium* mycotoxins, such as deoxynivalenol, T-2, HT-2, fumonisins, and zearalenone, whose presence in food and feed has been regulated by authorities, no limits have been set for ENN B, up to now.

#### 2.1.2. Tryptophol 

From all analyzed cheeses, the second most frequently detected metabolite was tryptophol. Tryptophol was detected in 30.8% of commercial cheeses (*n* = 12) and 34.5% of traditional cheeses (*n* = 10). Concentrations exhibited a broad range; in traditional cheese samples, the range varied from 13.4 to 7930 µg/kg (average concentration: 916 µg/kg), while the range in commercial cheese samples was between 15.5 and 354 µg/kg (average concentration: 136 µg/kg).

Tryptophol (indole-3-ethanol) is a metabolite produced by plants, bacteria, fungi and sponges. The central role of indolic compounds as plant growth regulators is well established [49]. The dairy yeast *Debaryomyces hansenii* was investigated for its production of alcohol-based quorum sensing (QS) molecules [50]. The addition of tryptophol was found to influence both the adhesion and sliding motility of this yeast. This fungus has great importance for the food industry; thus, it is used as a starter culture for the production of cheese products [51]. More recently, tryptophol was identified as a QS molecule released by cells and capable of inducing the morphogenetic switch in response to nitrogen starvation, which stimulates morphogenesis of pseudohyphal growth in *Sacchromyces cerevisiae* [52]. Tyrosol and tryptophol have been reported to impart slightly bitter flavors to beer. However, they have not been identified in cheese. The production of indolic compounds was screened, also, by rumen bacteria isolated from grazing ruminants. Fresh isolates from sheep and dairy cows produced indole, indole propionic acid, tryptophol and skatole from the fermentation of tryptophan and indoleacetic acid [53].

#### 2.1.3. 3-Nitropropionic Acid

Moreover, the metabolite 3-nitropropionic acid (3NPA) was found in one sample of Camembert cheese, at a concentration level of 66.7 µg/kg, and in one sample of traditional blue cheese Niva, at a concentration level of 10.6 µg/kg. In addition to 3NPA, all metabolites are confirmed, such as typical metabolites of *P. Roquefortii,* and their evidence in cheeses is not so strange and surprising. Our Camembert cheese sample (number 4) represented the white ripening Camembert cheese where *P. candidum* was used as a ripening culture. *P. Candidum* (synonym of *P. camemberti*) is a species domesticated from *P. commune*. This species has never been found outside the white mold cheese environment [54]. There is no evidence of the production of 3-NPA by this species. The neurotoxin 3-nitropropionic acid (3-NPA) is a mitochondrial toxin produced by several plants and fungi. The first production of 3-NPA (ß-nitropropionic acid) was derived from a strain of *A. flavus*. Since then, more fungi species have been added to producers of this mycotoxin: *A. wentii*, *A. oryzae*, *P. atrovenetum*, *Arthrinium* spp., *Mucor circinelloides* [55,56,57]. The consumption of 3-NPA by humans is likely through the consumption of both commercially and domestically prepared foodstuffs using fungi (*Penicillium* spp., *Aspergillus* spp.), or through forage consumption that includes contaminated plants [58]. Unfortunately, the survey of the natural contamination of 3-NPA is limited in the literature, although the presence of this metabolite was reported in cheese [10,59]. 

#### 2.1.4. Clavine Alkaloids, Isofumigaclavine, Festuclavine, and Chanoclavine

Among clavine alkaloids, isofumigaclavine, festuclavine, and chanoclavine were detected in our samples, especially in Niva cheeses. Chanoclavine and festuclavine varied relatively in low concentrations (3.1–7.5 µg/kg and 0.21–2 µg/kg, respectively; 5.9%, *n* = 4). Iso-fumigaclavine concentrations were higher (90–294 µg/kg). Ergot alkaloids are fungal metabolites with high biological activity, distinct in two subgroups. The first represents simple clavine-type alkaloids (for example, fumigaclavines) produced by fungi of the Aspergillaceae (*A. fumigatus*) [60], and the second includes lysergic acid-derived ergot alkaloids produced by parasitic or endophytic Clavicipitaceae fungi [61]. All three alkaloids were isolated from the collection and mutant strains of *P. roquefortii* [25]. *P. roqueforti* is able to accumulate the intermediates festuclavine and agroclavine [62]. *P. roqueforti* is used industrially as inoculum for ripening cheeses, and this saprophytic fungus produces a variety of enzymes and secondary metabolites (including mycotoxins) based on various types of substrates including, for example, PR-toxin, roquefortine C, isofumigaclavines (A, B), mycophenolic acid, andrastins (A, D), eremofortines (A, B, D) and others [27,63]. Fabian et al. [64] reported that the *P.*
*camemberti* genome contains a cluster of five genes required for the synthesis of the ergot alkaloids intermediate chanoclavine-I aldehyde. They analyzed samples of Brie and Camembert cheeses, as well as cultures of *P. camemberti*, and did not detect chanoclavine-I aldehyde or its derivatives. The production of festuclavine and iso-fumigaclavine was confirmed by *P. carneum* [62,65], which is included in the *P. roqueforti* complex. 

#### 2.1.5. Andrastins

Andrastins are interesting anticancer drug candidates which could represent an idea for the future production of “functionalized cheeses” with higher quantities of andrastin A [66]. Even though they are considered to be beneficial for human health, there are no studies that support its lack of toxicity when accumulated in high levels in cheeses. The first finding of andrastins in blue cheese was described by Nielsen et al. [27], where it was produced by the secondary starter culture *P. roqueforti*. In 23 representative samples of European blue cheeses, andrastin A was consistently found in quantities between 0.1 and 3.7 μg/g of mold-ripened cheese. Andrastin A was accumulated inside blue cheeses inoculated with this secondary starter. Not so far away, there was sequenced and annotated a genomic region that is involved in the biosynthesis of andrastin A in *P. roqueforti* [67]. Secondary metabolites produced by species of *P. roqueforti* complex (*P. roqueforti* and *P. paneum*) were established and two metabolites (roquefortine C and andrastin A) were consistently produced by both species [65]. A total of four white-mold-ripened cheeses were also purchased and used as controls; andrastins A-D were not detected. No correlation was observed between the level of sporulating mycelium (assessed visually) in the cheeses and the quantity of the andrastins [27]. This was expected, because they have never detected the production of andrastins by *P. camemberti* cultures [54]. Our investigation discovered the presence of andrastins (andrastin A, B, C and D) in 16 Slovak cheeses, traditional and commercial samples. Andrastin concentration levels ranged from 89.7 to 9140 μg/kg, with an average range of 3310 μg/kg (positive samples, *n* = 16). In our samples, andrastin A levels were the highest compared with others.

#### 2.1.6. Roquefortine C (ROQC), Roquefortine D (ROQD), Mycophenolic Acid (MPA)

*P. roqueforti* is highly appreciated in biotechnological applications and is a very important filamentous fungus, used during the process of making, and in the maturation of blue Roquefort-type cheeses. This fungus oxidizes fatty acids into methyl ketones, 2-haptanone and 2-honanone, which are considered to be responsible for the specific and unique sensorial flavors and odors of the blue cheese. On the other hand, there is increasing evidence that has been reported on the ability of this species to produce secondary metabolites in different culture media, or in the blue cheeses. This species is known as a producer of roquefortines, as well as of mycophenolic acid, amongst other mycotoxins [54,68]. Although ROQC and MPA present low acute cytotoxicity on the human intestinal cell, they have been shown to possess neurotoxic and immunosuppressive effects, respectively, and may thus cause secondary (indirect) mycotoxicosis [54]. Moreover, a high variability of these metabolites was observed in an unprecedented worldwide blue-veined cheese collection. Overall, 75% of samples contained less than 792 µg/kg ROQC and 705 µg/kg MPA [69]. Zambonin et al., [70] found MPA in five samples of Gongonzola and Danablu cheeses, ranging from 100 to 500 µg/kg. Another investigation [71] conducted on 53 blue cheeses reported that the levels of MPA achieved a concentration level from <10 to 1200 µg/kg. Our results show lower values of MPA range from 0.129 to 0.235 mg. ROQC and ROQD detected in the concentration levels varied from 592 to 17,900 µg/kg (positive sample, *n* = 8). Another study was conducted on industrial batches of nine *P. roqueforti* strains used in the production of the Gorgonzola cheese to verify the production of secondary metabolites. In vitro, only one *Penicillium spp* out of nine produced ROQC and four strains produced MPA. ROQC concentrations ranging from 50 to 1470 µg/kg were quantified after the analysis of 30 blue cheeses [72]. Moreover, higher values were obtained by Kokkonen et al. [24], from 600 to 12,000 µg/kg, respectively. There is a consensus that roquefortines in cheeses do not pose human health risks [62,68,69,73]. 

### 2.2. Method Performance Data

Method validation was performed in accordance with Sulyok et al. [31]. Table 2 reports the apparent recovery expressed as a percentage (%) and standard deviation (RSD %) for the 28 positively identified analytes in the investigated cheese samples. Apparent recovery was determined by adding a known quantity of a multi-analyte stock solution at a high concentration level to five individual blank samples.

## 3. Conclusions

The results of this current study indicate the presence of fungal mycotoxin contamination: ENN B was found in all samples (100%, *n* = 68), although its concentration was low compared with other metabolites (0.01 to 0.71 µg/kg), and tryptophol, the second most frequent detected metabolite, present in 30.8% of commercial cheeses (*n* = 12) and 34.5% of traditional cheeses (*n* = 10). Contamination with AFM1, the only metabolite regulated in EU milk and milk products, has not been confirmed. This is evidence of good milk quality and safety for human health. 

Continuing to monitor and describe the source and producers of contamination is essential. To protect consumers´ health, it is very important to verify and monitor the presence of secondary metabolites, including mycotoxins, in food that is widely consumed, to avoid mycotoxin risk. To our knowledge, this is the first preliminary investigation focused on Slovak cheeses and the contamination of these dairy products with mycotoxins.

## 4. Materials and Methods

### 4.1. Chemicals

Acetonitrile, methanol and glacial acetic were purchased from VWR Chemicals (Vienna, Austria). Ammonium acetate was acquired from Sigma-Aldrich (Vienna, Austria). A Purelab Ultra system (ELGA LabWater, Celle, Germany) was used for further purification of reverse osmosis water. Standards of fungal metabolites were purchased from several sources as described in [31].

### 4.2. Sampling

Altogether, 68 samples of cheese were randomly purchased from food markets, as well as directly from local cheesemakers located in the south-western part of Slovakia. All cheese samples originated from 3 types of Slovak milk: cow, sheep, goat. Our sampling can be divided into 2 types of cheeses: common commercial Slovak cheeses came from markets (*n* = 39, Appendix A), and original traditional Slovak cheeses came from markets and local farmers using a traditional biotechnical process at manufacture (*n* = 29, Appendix A). Traditional cheeses represent unique and specific types of cheese not found anywhere else in the world. We talk about traditional sheep cheese, Bryndza, Camembert cheese, Encián, Plesnivec, and blue cheese, Niva, from cow’s milk. Camembert white brine cheese is made by a culture of *P.candidum,* while Niva, as an internally mold-ripening cheese, is formed by *P. roquefortii*. Other origin types of cheeses were korbáčik-traditional Slovak, Nite traditional cheese threads, Parenica and Oštiepok, all derived from cow’s milk. 

### 4.3. Mycotoxin Extraction 

Mycotoxin extraction involved the addition of 20 mL of acetonitrile/water/acetic acid (79:20:1, *v*/*v*/*v*) to 5 g of homogenized cheese samples according to the procedure described by [31]. The mixture was placed on a rotary shaker (GFL 3017, Burgwedel, Germany) for 90 min. After centrifugation for 5 min at 5000 rpm the supernatant was diluted two times with acetonitrile/water/acetic acid (20:79:1, *v*/*v*/*v*) and analyzed through mass spectrometry.

### 4.4. LC-ESI-MS/MS Analysis

The analytical conditions used for the analysis of mycotoxins are reported by [31]. Analyses were performed using a QTrap 5500 MS/ MS (Sciex, Foster City, CA, USA) equipped with a TurboIonSpray electrospray ionization (ESI) source and a 1290 Series HPLC System (Agilent, Waldbronn, Germany). Chromatographic separation was carried out by using a Gemini C18 column, 150 × 4.6 mm, 5 μm particle size, equipped with a security guard cartridge (Phenomenex, Torrance, CA, USA) held at 25 °C. The mobile phase consisted of (A) methanol:water (10:90 *v*/*v*) and (B) methanol:water (98:2 *v*/*v*), both containing 5 mM ammonium acetate and 1% acetic acid. Elution was carried out as follows: an initial 100% A was held for 2 min and decreased to 50% A over 3 min. Then, the gradient was linearly decreased to 0% A over 9 min, followed by a hold time of 4 min at 0% A and held for 2.5 min for re-equilibration. The flow rate was 1 mL/min. The injection volume was set to 5 μL. ESI MS/ MS was performed both in positive and negative polarity. The target cycle time was set to 1000 ms, the MS pause time at 3 ms, and the detection window width was 52 and 40 s in the negative and positive ESI mode, respectively. For confirmation criteria, the ion ratio was compared with the related value of the standard at a tolerance within 30%, and the retention time at a tolerance of ±0.03 min.

### 4.5. Validation Method

External quantification was performed using a serial dilution of a multi-analyte stock solution. Results were corrected for apparent recoveries that were determined through spiking experiments. The limits of detection and quantification were determined following the EURACHEM guide [31]. The accuracy of the method is verified on a routine basis by participation in a profinite testing scheme organized by BIPEA (Geneviliers, France).

### 4.6. Statistical Analysis

Statistical analysis was performed by using the software Info-Stat (https://www.infostat.com.ar/index.php?mod=page&id=15 accessed on 27 January 2022; Córdoba, Spain), version 2008. Tukey’s test was used to assess differences between the various typologies of the studied samples. Tukey’s test was considered significant at a *p* < 0.05 level. Three independent replications were used to assess the results, expressed as mean ± standard deviation (SD).

## Figures and Tables

**Table 1 toxins-14-00134-t001:** Summary of the analyzed cheeses (*n* = 68) and their concentration ranges expressed as µg/kg.

Type of Mycotoxin	Cow’s Cheeses/45 Samples			Sheep’s Cheeses/19 Samples			Goat’s Cheeses/4 Samples			
	Commercial	Traditional	Concentration Range µg/kg	Commercial	Traditional	Concentration Range µg/kg	Commercial	Traditional	Concentration Range µg/kg	Numbers of Total Positive Samples
3-NPA *	0/25	2/20	10.6–66.7	3/10	0/9	3.8–27.2	0/4	-	-	5/68
Andrastin A, B, C, D	0/25	8/20	89.7–8890	8/10	0/9	96.7–9140	0/4	-	-	16/68
Chanoclavine	0/25	2/20	3.1–7.5	2/10	0/9	4.8–6.9	0/4	-	-	4/68
Enniatin B	25/25	20/20	0.02–0.71	10/10	9/9	0.04–0.57	4/4	-	0.09–0.17	4/68
Festuclavine	0/25	2/20	1.2–2	2/10	0/9	0.21–1.3	0/4	-	-	4/68
iso-Fumigaclavine	0/25	2/20	178–294	2/10	0/9	90–135	0/4	-	-	4/68
Mycophenolic acid	0/25	2/20	20.7–29.4	2/10	0/9	16.1–28.7	0/4	-	-	4/68
Roquefortine C, D	0/25	4/20	591.8–17900	4/10	0/9	679–13700	0/4	-	-	8/68
Tryptophol	8/25	8/20	15–7930	4/10	2/9	13.4–171	0/4	-	-	22/68

***** 3-Nitropropionic acid.

**Table 2 toxins-14-00134-t002:** Apparent recovery expressed as percentage (%) and standard deviation (RSD) for the 28 positively identified analytes in Slovak cheese samples.

Analyte	Apparent Recovery (%)	RSD (%)
3-Nitropropionic acid	53.2	14.9
Aflatoxin B1	28.3	10.4
Aflatoxin M1	50.7	13.9
Alternariolmethylether	75.1	13.9
Andrastin A	54.9	18.2
Andrastin B	59.5	12.7
Andrastin C	60.8	7.4
Chanoclavine	39.8	19.2
Citreoviridin	84.1	16.2
Citrinin	107	13.5
Cyclopiazonic acid	102.9	13.9
Deoxynivalenol	77.8	9
Enniatin B	93	7.2
Fumigaclavine A	48	15.9
Fumonisin B1	61.2	17.5
Fumonisin B2	72.3	15.5
Griseofulvin	49.1	10.4
Mycophenolic acid	71.6	14.2
Ochratoxin A	68.7	9.3
Patulin	47.8	18.9
Penitrem A	116.8	12.8
Quinolactacin A	43.5	17.2
Roquefortine C	71.7	12.2
Roquefortine D	36.3	13.3
Sterigmatocystin	61	18.1
T-2 Toxin	56.3	5.6
Tryptophol	31.8	9.6
Zearalenone	71.2	9.1

## Data Availability

Not applicable.

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
