# Peer review of "Analysis of Mycotoxin and Secondary Metabolites in Commercial and Traditional Slovak Cheese Samples"

_toxins, 2022, doi:10.3390/toxins14020134_

Round 1
Reviewer 1 Report
Manuscript ID: toxins-1595925: Analysis of mycotoxin and secondary metabolites in commercial and traditional Slovak cheese samples
The manuscript describes the content of mycotoxins and secondary metabolites of fungi in commercial and traditional Slovak cheese.
The manuscript is interesting and well written and documented. However, some comments and suggestions are addressed to the authors:
Among all the samples, only 4 samples (samples: 4, 6, 30 and 37) contained almost all the measured target compounds. in addition, when we compare samples 30 and 37, which are commercial sheep cheeses, respectively not smoked and smoked, we observe not big difference between their compounds contents. This aspect is not commented in the manuscript.
L62-63: please an example of work studying the control of cheese contamination by fungi.
The concentration ranges of ENN B indicated in L154 (0.06 to 0.17 µg/kg) and in L156 (0.21 and 0.15 µg/kg) don’t correspond to the limits observed in Table 1. Please verify that.
L186: the maximal content of tryptophol in samples is 7930,4 not 7960 ug/kg. Please see Table 1 (sample 3).
L285: 17858 µg/kg or 17900 µg/kg which is indicated in Table 1. Please verify that.
Author Response
Response to reviewer 1
Manuscript ID: 1595925
Title: Analysis of mycotoxin and secondary metabolites in commercial and traditional Slovak cheese samples
Reviewer: 1
The manuscript describes the content of mycotoxins and secondary metabolites of fungi in commercial and traditional Slovak cheese. The manuscript is interesting and well written and documented. However, some c796omments and suggestions are addressed to the authors:
Point 1: Among all the samples, only 4 samples (samples: 4, 6, 30 and 37) contained almost all the measured target compounds. in addition, when we compare samples 30 and 37, which are commercial sheep cheeses, respectively not smoked and smoked, we observe not big difference between their compounds contents. This aspect is not commented in the manuscript.
Response 1: As suggested by reviewer 1, the authors added this missing information in the manuscript.
Point 2: L62-63: please an example of work studying the control of cheese contamination by fungi.
Response 2: As suggested by reviewer 1, the authors reported an example about the control of cheese contamination by fungi.
Point 3: The concentration ranges of ENN B indicated in L154 (0.06 to 0.17 µg/kg) and in L156 (0.21 and 0.15 µg/kg) don’t correspond to the limits observed in Table 1. Please verify that.
Response 3: As suggested by reviewer 1, the authors checked these values in the manuscript.
Point 4: L186: the maximal content of tryptophol in samples is 7930,4 not 7960 ug/kg. Please see Table 1 (sample 3).
Response 4: As suggested by reviewer 1, the authors modified the value.
Point 5: L285: 17858 µg/kg or 17900 µg/kg which is indicated in Table 1. Please verify that.
Response 5: As suggested by reviewer 1, the authors modified the value.
The authors thank reviewer 1 for the precious suggestions helpful in improving the manuscript.
Reviewer 2 Report
- There are few corrections in the manuscript which have been highlighted in the yellow colour and suggestions given in the pop up note.
- The sampling methodology and the sampling strategy followed in the selection of 68 and its geographical location may be explained in the materials and methods section.
- The table no. 1 and 4 may be given in the supplementary files rather than in the manuscript.
- Conclusions may be improved and should be based on the present study.

Author Response
Response to reviewer 2
Manuscript ID: 1595925
Title: Analysis of mycotoxin and secondary metabolites in commercial and traditional Slovak cheese samples
Reviewer: 2
Manuscript ID: toxins-1595925: Analysis of mycotoxin and secondary metabolites in commercial and traditional Slovak cheese samples
Point 1: There are few corrections in the manuscript which have been highlighted in the yellow colour and suggestions given in the pop up note.
Response 1: As suggested by reviewer 2, the authors made the suggested changes.
Point 2: The sampling methodology and the sampling strategy followed in the selection of 68 and its geographical location may be explained in the materials and methods section.
Response 2: As suggested by reviewer 2, the authors added the missing information in the manuscript.
Point 3: The table no. 1 and 4 may be given in the supplementary files rather than in the manuscript.
Response 3: As suggested by reviewer 2, the authors moved table 1 and 4 in the supplementary section.
Point 4: Conclusions may be improved and should be based on the present study.
Response 4: As suggested by reviewer 2, the authors improved the conclusion section.
The authors thank reviewer 2 for the precious suggestions helpful in improving the manuscript.